# Normoferremia in Patients with Acute Bacterial Infections—A Hitherto Unexplored Field of the Dichotomy between CRP and Ferritin Expression in Patients with Hyper Inflammation and Failure to Increase Ferritin

**DOI:** 10.3390/ijms241411350

**Published:** 2023-07-12

**Authors:** Tal Levinson, Eugene Feigin, Shlomo Berliner, Shani Shenhar-Tsarfaty, Itzhak Shapira, Ori Rogowski, David Zeltzer, Ilana Goldiner, Moshe Shtark, Malka Katz Shalhav, Asaf Wasserman

**Affiliations:** 1Infectious Diseases Unit, Tel-Aviv Sourasky Medical Center Affiliated to the Faculty of Medicine, Tel-Aviv University, Tel Aviv 6423906, Israel; 2Departments of Internal Medicine C, D and E, Tel Aviv Sourasky Medical Center Affiliated to the Faculty of Medicine, Tel Aviv University, Tel Aviv 6423906, Israel; eugenef@tlvmc.gov.il (E.F.); berliners@tlvmc.gov.il (S.B.); shanis@tlvmc.gov.il (S.S.-T.); shapira@tlvmc.gov.il (I.S.); orir@tlvmc.gov.il (O.R.); asafw@tlvmc.gov.il (A.W.); 3Department of Endocrinology, Tel Aviv Sourasky Medical Center Affiliated to the Faculty of Medicine, Tel Aviv University, Tel Aviv 6423906, Israel; 4Department of Emergency Medicine, Tel-Aviv Sourasky Medical Center, Affiliated to the Faculty of Medicine, Tel Aviv University, Tel Aviv 6423906, Israel; davidz@tlvmc.gov.il (D.Z.); malkak@tlvmc.gov.il (M.K.S.); 5Clinical Laboratory Services, Tel Aviv Sourasky Medical Center, Tel Aviv, Affiliated to the Faculty of Medicine, Tel Aviv University, Tel Aviv 6423906, Israel; ilanag@tlvmc.gov.il (I.G.); moshes@tlvmc.gov.il (M.S.)

**Keywords:** acute phase response, C-reactive protein, ferritin, hyperinflammation

## Abstract

Ferritin is an acute phase response protein, which may not rise as expected in acute bacterial infections. This could be due to the time required for its production or to a lack of response of ferritin to the bacterial inflammatory process. Medical records of hospitalized patients with acute hyper inflammation were retrieved and studied, looking closely at two acute phase proteins: C-reactive protein (CRP) and ferritin. The estimated time between symptom onset and the procurement of blood tests was also measured. 225 patients had a median ferritin level of 109.9 ng/mL [IQR 85.1, 131.7] and a median CRP level of 248.4 mg/L [IQR 221, 277.5]. An infectious inflammatory process was identified in 195 patients. Ferritin levels were relatively low in comparison with the CRP in each group, divided according to time from symptom onset until the procurement of blood tests. The discrepancy between high CRP and low ferritin suggests that these two acute phase response proteins utilize different pathways, resulting in a failure to increase ferritin concentrations in a documented state of hyperinflammation. A new entity of normoferremic inflammation accounts for a significant percentage of patients with acute bacterial infections, which enables bacteria to better survive the inflammation and serves as a new “inflammatory stamp”.

## 1. Introduction

It is well documented that ferritin is an acute phase response protein that increases during inflammation [1]. However, it has been suggested that it may not be raised in patients with acute bacterial infections [2].

During an episode of bacterial infection, the bacteria and the host must compete for essential nutrients. One of the most important nutrients is iron, due to its role in a wide range of enzymatic processes including energy metabolism, DNA synthesis and reactive oxygen species defense [3]. Due to the poor aqueous solubility of ferric ion (Fe^3+^) in neutral pH and in the presence of oxygen, free iron is not found in the host but is instead sequestered in complexes of proteins able to bind iron, such as lactoferrin, transferrin, protoporphyrins and ferritin [4].

The host aims to deprive the bacteria of iron, while the bacteria attempts to overcome these mechanisms of iron depletion. However, iron may have deleterious consequences for the host due to its ability to produce reactive oxygen species. It is therefore stored intracellularly in ferritin to create a relatively safe environment for iron storage by detoxifying iron and preventing the formation of reactive oxygen species. The uptake of iron by ferritin commences with the oxidation of Fe^2+^ by molecular oxygen at the ferroxidase center, and iron is thus stored as Fe^3+^. When necessary, the ferritin molecule releases its stored iron while keeping the iron in a nonreactive state, providing a protective antioxidant function [5].

Ferritin consists of two types of subunits, the ferritin L-chain (FTL) and the ferritin H-chain (FTH), which assemble in variable ratios to form a heteropolymeric protein [6]. Different organs and tissues will contain ferritin rich either in FTL or in FTH. It is presumed that FTL-rich tissues are more involved in the storage of iron (for example, the liver and spleen) and that FTH-rich tissues tend to serve more anti-oxidative functions (for example, in the muscle and kidney) [7]. Moreover, the ferritin subunits are active in immunomodulation. The FTH subunit has been shown to possess not only pro-inflammatory but anti-inflammatory properties, too, depending on the cell type [8,9]. Additionally, ferritin has immunosuppressive effects. Apoferritin, which is the iron-free protein, was demonstrated to mitigate LPS-mediated macrophage activation [10]. Haschka et al. examined mice lacking FTH in the myeloid lineage and found that these mice displayed impaired iron storage capacities in the tissue’s leukocyte compartment, as well as increased levels of iron in macrophages and an accelerated macrophage-mediated iron turnover. It was shown that the depletion of FTH evoked an impaired control of Salmonella infection in mice displaying increased bacterial burden, the uncontrolled activation of NF-κB and inflammatory signaling, the development of cytokine storm and subsequent death [11].

It is known that the production of pro-inflammatory cytokines and acute phase response proteins affects iron metabolism. For example, hepcidin production is increased in the acute phase response to infection, leading to the decreased export of iron, making it less available to infectious bacteria [12]. Moreover, neutrophil gelatinase-associated lipocalin (NGAL) inhibits siderophore-dependent iron uptake, compromising a central mechanism that bacteria use to acquire iron [13,14]. Furthermore, lactoferrin has a high affinity for iron and binds it at the mucosal level, thus withholding iron from the pathogen.

We hypothesized that one of the reasons responsible for ferritin not being elevated in patients with acute bacterial infections could be a time lag between disease onset and the increased production of ferritin.

In order to find out whether ferritin is indeed non-responsive to the bacterial inflammatory process or whether there is simply a delay in its production, we performed this study, looking specifically at the correlation between time from symptom onset and ferritin concentration in a group of patients with hyperinflammation.

This study was carried out under the assumption that a relatively large proportion of hospitalized patients with very high concentrations of C-reactive protein (CRP) concentrations had an acute inflammatory pathology, whether infectious or non-infectious. Our model therefore examined a unique patient group who presented with hyperinflammation but had no significant rise in their concentration of ferritin, henceforth defined as normoferremia.

The principal aim of this study was to understand whether normoferremia during acute infection/inflammation was a result of this protein being simply a late acute phase response biomarker, or whether in fact we are seeing evidence of an inverse relationship between CRP and ferritin in certain clinical scenarios.

## 2. Results

A total of 225 patients were included in this cohort. Fifty-four of them were male and one hundred and seventy-one were female, with a median age of 74.1 [IQR 59.9, 85.7] and 72.5 years [IQR 48.9, 82.9], respectively. The median ferritin level was 109.9 ng/mL [IQR 85.1, 131.7] and the median CRP level was 248.4 mg/L [IQR 221, 277.5]. A total of 195 patients had infectious etiologies for inflammation, while 18 had non-infectious etiologies and 12 had undetermined etiologies for inflammation. Pneumonia, urinary tract infections and skin infections were the leading infectious diagnoses (38.46%, 30.77% and 11.28%, accordingly) (Table 1). In eighty-nine of the infected patients, the causative agent was identified. E. coli was found to be the leading pathogen (Table 2).

Dividing the cohort into groups according to the estimated time between symptom onset and the procurement of blood tests, twenty-eight patients arrived within 24 h, forty-nine patients between 24.1 and 48 h, forty patients between 48.1 and 72 h, forty-three patients between 72.1 and 96 h and sixty five arrived later than 96 h after symptom onset. Their characteristics are described in Table 3. Figure 1 shows the median ferritin and CRP values of the cohort according to the estimated time from symptom onset to presentation. Among those with elevated CRP, suggestive of an underlying inflammatory process, Figure 1 demonstrates that ferritin levels are comparatively low across all groups. There was no statistical difference either in ferritin values or in CRP values between the different time groups. The number of cases caused by an infectious etiology varied in each of these groups: twenty-four cases within 24 h, forty-five cases between 24.1 and 48 h, thirty-five patients between 48.1 and 72 h, thirty-nine patients between 72.1 and 96 h and fifty-two arrived later than 96 h after symptom onset. Contrastingly, the time between symptom onset and presentation in cases caused by non-infectious etiology was similar among most groups: four cases within 24 h, two cases between 24.1 and 48 h, four patients between 48.1 and 72 h, four patients between 72.1 and 96 h and four arriving later than 96 h after symptom onset. The comparison of inflammatory etiologies according to the estimated time from symptom onset to presentation is shown in Table 4.

## 3. Materials and Methods

### 3.1. The Patients

We included patients who were hospitalized in the Tel-Aviv Sourasky Medical Center, a tertiary university-affiliated 1170-bed acute care hospital located in the center of Israel, who had acute hyper inflammation arbitrarily defined as CRP concentrations above 200 mg/L (mega CRP). The data were obtained using MDClone (mdclone.com, accessed on 1 September 2022), a query tool that provides comprehensive patient-level data of wide-ranging variables in a defined timeframe around an index event. The MDClone system is a patient-level data extraction system designed to ease the analysis of large medical records. Two senior physicians (S.B. and T.L.) reviewed the medical records of all included patients to estimate the time elapsed (in hours) between the onset of signs and symptoms and the measurement of two acute phase proteins, namely CRP and ferritin, obtained in a time frame of no more than 6 h apart. When referring to signs and symptoms of disease activity, we included typical clinical presentations of inflammation such as fever, shivering, cough, flank pain, urinary symptoms, etc.

### 3.2. Laboratory Methods

The ADVIA Centaur Ferritin assay is a two-site sandwich immunoassay using direct chemiluminometric technology, which uses constant amounts of two anti-ferritin antibodies. The first antibody, in the Lite Reagent, is a polyclonal goat anti-ferritin antibody labeled with acridinium ester. The second antibody, in the Solid Phase, is a monoclonal mouse anti-ferritin antibody, which is covalently coupled to paramagnetic particles. At our laboratory, the normal ranges of ferritin values are 14–163 ng/mL and 7.1–151 ng/mL for men and women, respectively. We therefore assigned 151 ng/mL as the upper limit of normal in order to include both normoferremic men and women in our study.

wrCRP measurements were performed by ADVIA^®^ (Siemens Healthcare Diagnostics Inc., Tarrytown, NY, USA). The ADVIA^®^ Chemistry wrCRP method measures CRP in blood using a latex-enhanced immunoturbidimetric assay. It is based on the principle that the analytic concentration is a function of the intensity of scattered light caused by the latex aggregates. Latex particles coated with anti-CRP antibodies rapidly agglutinate in the presence of CRP-forming aggregates. Agglutination takes place with increases in turbidity, which was measured at 571 nm. This method measures the wrCRP concentration range of 0.03–155 mg/L. When the measured concentrations exceeded 155 mg/L, a dilution of 1:4 was performed to extend this range.

### 3.3. Statistical Methods

Categorical variables were described using frequency and percentage. Continuous variables were evaluated for normal distribution using histograms and Q-Q plots and reported as median and interquartile range (IQR). A chi-square test was used to compare categorical variables between different time groups. All statistical tests were two tailed, and *p* < 0.05 was considered statistically significant. All statistical analyses were performed using SPSS (IBM SPSS Statistics for Windows, version 27, IBM Corp., Armonk, NY, USA, 2020).

## 4. Discussion

To the best of our knowledge, this is the first study to describe the principal etiologies of hyperinflammation amongst patients who present at the hospital with normoferremia. The main finding of this study was that the discrepancy between the high CRP and low ferritin observed is not a result of different time delays between disease onset and presentation. Rather, it is a real biological phenomenon highlighting the different pathways of these two acute phase response proteins and the failure to increase ferritin concentrations in a documented state of hyperinflammation. A different, though opposite, response of these two biomarkers has been recently reported in patients with hemophagocytic lymphohistiocystosis (HLH), in whom hyperferritinemia has been observed in the presence of relatively low CRP concentrations [15]. Such a dichotomy might be clinically useful as these findings could be revealing a unique “inflammatory stamp”. This study describes the opposite situation, in which normoferremia exists in patients with very high concentrations of CRP, which confirms the presence of a significant inflammatory response. Furthermore, ferritin regulation is influenced by multiple factors involved in the inflammatory cascade: the inflammatory stimulus of reactive oxygen species, pro-inflammatory cytokines tumor necrosis factor α, IFN-γ, IL-6 and the anti-inflammatory IL-10. These all contribute to ferritin expression.

Little is known about the time course of ferritin concentration changes in patients suffering from an acute inflammatory response to various noxious stimuli. One study was performed in patients who underwent heart surgery, and reported a significant rise in ferritin concentrations 96–144 h postoperatively [16]. However, only seven patients were included, and by looking at the graphical presentation one could observe that a rise in ferritin concentrations can be detected as early as one day following surgery.

The effect of fever on serum ferritin was investigated by Elin et al., who reported that the ferritin response following the induction of fever by either endotoxin or etiocholanole in healthy volunteers reached its maximum level at 3–4 days [17]. Again, it was documented that a rise in serum ferritin could be detected as early as 24 h. In another study, reported by Birgegard et. al. [18], it was shown that the ferritin elevation started within two days of the onset of fever in a group of eight transplant patients with acute infections. Following adaptive immunotherapy with chimeric antigen receptor T (CART) cells, the maximal ferritin concentration was seen on day 4 [19]. These reports clearly demonstrate that ferritin levels can be expected to reach their peak within 3–4 days following inflammatory stimuli. This suggests that in our study we witnessed a failure to produce hyperferritinemia in the presence of acute infections that have caused a hyperinflammatory response.

It is well documented that acute infections can be associated with hyperferritinemia [20], including both bacterial and viral etiologies [21,22]. Therefore, our study may define a new entity of normoferremic inflammation and suggests that a high percentage of patients with acute infections, especially bacterial ones, will follow this pattern. The availability of free iron, the so-called nutritional immunity [23], could present survival advantages for bacteria, as discussed in several clinical scenarios previously [24,25,26,27,28,29,30,31]. Bacterial virulence and ability to proliferate are related to the availability of free iron in their environment [32], and in order to establish the infection the bacteria must acquire the host’s iron. Indeed, iron overload is associated with an increased risk of infection.

Muhsen et al. studied the association between Helicobacter pylori seropositivity and ferritin levels among Israeli-Arab children, a population with a high prevalence of Helicobacter pylori infection and high anemia rates [33]. The study examined the children’s serum samples from 2000 to 2001, measuring serum Helicobacter pylori-specific IgG antibodies. A total of 509 serum samples were included, with a Helicobacter pylori seropositivity of 47.3% and low ferritin levels (defined as ferritin less than 12 μg/L) found among 11.4% of subjects. The authors concluded that in Helicobacter pylori seropositive children, 14.5% had low ferritin levels, compared to 8.6% among Helicobacter pylori seronegative children. This was a dose-response relationship between low ferritin and anti-Helicobacter pylori serum IgG levels. Interestingly, multivariate analysis demonstrated that, in children aged 5 years or younger, Helicobacter pylori infection was associated with low ferritin levels. The authors concluded that Helicobacter pylori acquired in early childhood could compete with the host for iron and deplete iron stores. Higher serum IgG levels may reflect a more severe gastric infection and a more significant reduction in serum ferritin levels. Possible mechanisms are gastrointestinal blood loss through breaks in the gastric mucosa, the decreased absorption of dietary iron and an enhanced uptake of iron by Helicobacter pylori bacteria. Additionally, a trend of low ferritin levels in Helicobacter pylori- CagA seropositive subjects was found, compared to Helicobacter pylori seropositive subjects lacking CagA antibodies and Helicobacter pylori seronegative children. It has been shown that CagA-positive Helicobacter pylori strains are associated with an intensity of gastritis and duodenal ulcers, suggesting a more severe disease. The clinical implications of the described study are yet to be confirmed, as anti-Helicobacter pylori treatment in children was not found to be superior over iron supplementation in resolving iron deficiency anemia in highly endemic populations for Helicobacter pylori [34].

Mycobacterium tuberculosis (MTB) represses iron acquisition and induces iron storage by two iron storage proteins, a ferritin (bfrB) and a bacterioferritin (bfrA). MTB upregulates the bfrA and bfrB genes under conditions of iron sufficiency. Pandey et al. found that bfrB is required to overcome iron limitation and for protection against oxidative stress, while bfrA was dispensable for adaptation to those stresses. MTB mutants lacking bfrB grew slower than wild type MTB and were eliminated from infected lungs of mice, suggesting that bfrB is necessary to survive the stress induced by the adaptive immune response as well as to establish a chronic infection. Moreover, a lack of bfrB rendered MTB sensitive to high iron levels and to iron-activated antibiotics such as Isoniazid, indicating that bfrB is essential to prevent iron toxicity and to maintain the iron homeostasis of MTB [35]. These findings can potentially improve the clinical outcome of MTB infection by developing a strategy of using antibiotic therapy combined with the inhibition of bfrB in order to potentiate the killing of MTB by antibiotics.

Vibrio vulnificus is a Gram-negative marine bacterium capable of causing a life-threatening sepsis syndrome with high mortality rates. Known risk factors associated with the host’s susceptibility to infection with Vibrio vulnificus are iron overload and hemochromatosis. Hor et al. demonstrated that the survival of Vibrio vulnificus in the blood was correlated with high serum ferritin levels and transferrin iron saturation. The authors could not determine whether ferritin enhances bacterial growth or if the ferritin level correlates with hepatic necroinflammation—another risk factor for Vibrio vulnificus infection [36].

Even though ferritin is traditionally regarded as a mechanism of the host for withholding iron from infecting bacteria, several studies have demonstrated the ability of certain bacteria to use ferritin as a source of iron. Neisseria meningitides triggers an iron starvation response and ferritinophagy, which provides a source of iron [37]. Several mechanisms were proposed that enable bacteria to use ferritin to acquire iron. Bacterial proteases mobilize iron in Pseudomonas aeruginosa and Burkholderia cenocepacia [38,39]. Uropathogenic Escherichia coli can persist in autophagosomes inside urothelial cells via ferritinophagy, allowing the bacteria access to iron [40]. Bacillus cereus can bind ferritin to its surface and acquire the iron from bound ferritin [41]. Listeria monocytognes uses surface-associated ferrireductases to acquire iron from ferritin [42]. A possible advantage of normoferremia in hosts infected with the above-described bacteria might be to reduce their ability to acquire iron from ferritin; however, this assumption should be further studied.

Physicians examine ferritin concentration as part of anemia investigation. In fact, in the present study, most patients had their iron and transferrin saturation measured concomitantly to ferritin. As shown in Table 3, anemia was relatively prevalent in this cohort.

The study cohort included a relatively high percentage of elderly women. Again, it is known that anemia is relatively prevalent in the elderly, and anemia itself was probably the reason the physicians tested for ferritin, iron and transferrin saturation. Another possibility is the relatively large number of patients with acute pyelonephritis, as well as other forms of urinary tract infections, which are known to exist in elderly women [43]. In the present study, we noted few patients with COVID-19, probably due to the fact the hyperferritinemia is generally found in patients with COVID-19, who are only rarely normoferremic [44].

There are several limitations of our present study, the main one being its retrospective nature. An additional limitation is the fact that serum ferritin is not a routine test performed in patients who are admitted with acute infections, but is probably used as part of the routine evaluation of patients with anemia. In fact, in most of our patients, serum iron, transferrin and transferrin concentrations were obtained in addition to serum ferritin. Moreover, we did not have access to the post-inflammatory values of ferritin and CRP of the same patients to verify the true state of their ferritin. It is likely that infections are highly prevalent in most patients who are admitted to the hospital with a hyperinflammatory response. Despite all of these limitations, our study is the first one to document significant discrepant pathways between CRP and ferritin in clinical practice. It is also the first to confirm that this newly described phenomenon of normoferremic inflammation results from a failure to increase the ferritin level and not due to ferritin being a relatively late acute phase reactant, as opposed to earlier ones like CRP or serum amyloid A (SAA) [45]. The biological impact, if any, of this newly defined entity of normoferremic inflammation is yet to be explored.

It can be concluded that normoferremic inflammation might present a new “inflammatory stamp” that could help explore the different pathways of inflammatory responses to noxious signals. The primary finding of this study is the confirmation that normoferremia during hyperinflammatory responses is not necessarily an incidental finding. Instead, it is a true failure to increase the concentration of this particular protein and not due to its nature of being a slower acute phase reactant compared to C-reactive protein.

## Figures and Tables

**Figure 1 ijms-24-11350-f001:**
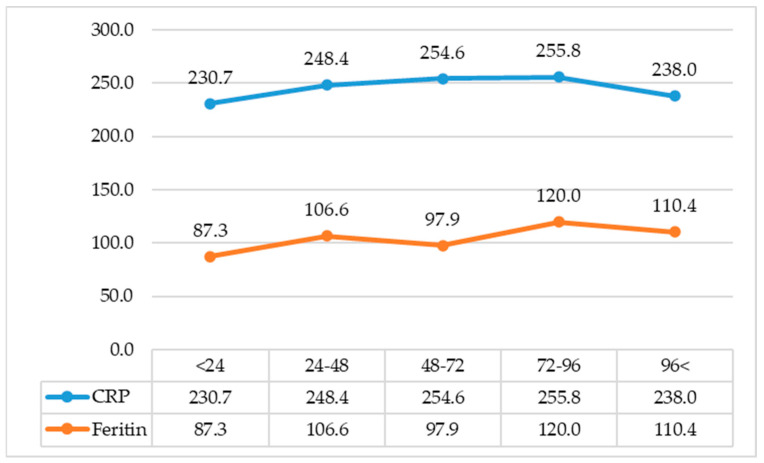
Mean CRP and ferritin values between groups by time from symptom onset to presentation. CRP—C-reactive protein.

**Table 1 ijms-24-11350-t001:** Infectious syndrome diagnosis among 195 infected patients.

Infectious Syndrome	No. of Patients (%)
Pneumonia	75 (38.46)
UTI	60 (30.77)
Skin infection	22 (11.28)
URTI	7 (3.59)
GI infection	7 (3.59)
COVID-19	7 (3.59)
Sepsis of unknown source	6 (3.08)
Nervous system infection	3 (1.54)
Other	8 (4.1)

COVID-19—corona virus disease 2019, GI—gastrointestinal, URTI—upper respiratory tract infection, UTI—urinary tract infection.

**Table 2 ijms-24-11350-t002:** Identified infectious agent diagnosis among 89 patients.

Infectious Agent		No. of Cases (%)
Bacteria	*E. coli*	41 (46.07)
	*Streptococcal* species	7 (7.87)
	*Haemophilus influenzae*	6 (6.74)
	*Klebsiella pneumoniae*	4 (4.49)
	*Proteus mirabilis*	3 (3.37)
	*Pseudomonas aeruginosa*	3 (3.37)
	*Enterococcal* species	2 (2.24)
	Other	9 (10.11)
Virus	SARS-CoV-2	5 (5.62)
	Influenza A	1 (1.12)
Mixed infection		8 (8.99)

SARS-CoV-2—severe acute respiratory syndrome corona virus 2.

**Table 3 ijms-24-11350-t003:** Patient characteristics, inflammatory biomarker level and type of inflammation divided into groups according to time (in hours) from symptom onset to the procurement of blood tests.

Time (h)		<24	24–48	48–72	72–96	>96	Total	*p* Value
		N = 28	N = 49	N = 40	N = 43	N = 65	N = 225	
Sex	Male	11.0	13.0	7.0	8.0	15.0	54.0	0.248
	Female	17.0	36.0	33.0	35.0	50.0	171.0	
Age (years)		73 (63.3–83.5)	73.3 (53.3–83.6)	76.6 (56.6–84.6)	74.9 (53.7–83.8)	66.1 (40.4–82.2)	73.3 (52.9–83.4)	0.161
Hemoglobin (g/dL)		10.1 (9.3–12.2)	10.2 (9.3–11.4)	10 (8.9–10.9)	10.1 (9.55–11.25)	9.7 (9–11.6)	10 (9.1–11.4)	0.744
Iron (mcg/dL)		9 (7–12.5)	8 (6–10.7)	8 (6.95–12)	10 (8.25–14)	11 (8–15.5)	10 (7–13)	0.671
Transferrin (mg/dL)		216 (181–226)	190 (152–214)	215 (184–250)	193 (169 –203)	196 (175–225)	196 (171 – 225)	0.247
Transferrin saturation (%)		3.2 (2–3.9)	3.2 (2.6–4.7)	2.8 (2.2–3.9)	4 (2.8–5.5)	4.1 (2.7–6.2)	3.3 (2.5–5.2)	0.669
Ferritin (ng/mL)		87.3 (7.3–122.9)	106.6 (86.9–135)	97.9 (83.4–128.9)	120 (97.5–141)	110.4 (86.8–133.7)	109.9 (85.1–131.7)	0.51
CRP (mg/L)		230.7 (218.7–268.7)	248.4 (215.5–301.1)	254.6 (223.4–276)	255.8 (240.6–285.9)	255.8 (240.6–285.9)	248.4 (221–277.5)	0.688
Type	Infectious	24	45	35	39	52	195	0.032
	Non-infectious	4	2	4	4	4	18	
	Undetermined	0	2	1	0	9	12	

CRP—C-reactive protein.

**Table 4 ijms-24-11350-t004:** Inflammatory syndrome comparison between different time groups from symptom onset to presentation.

	≤24 h	24.1–48 h	48.1–72 h	72.1–96 h	>96 h	Total
Infectious	24	45	35	39	52	195
Non-infectious	4	2	4	4	4	18
Undetermined	0	2	1	0	9	12
Total	28	49	40	43	65	225

## Data Availability

Data can be provided by the corresponding author upon reasonable request.

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
