# Peer review of "Normoferremia in Patients with Acute Bacterial Infections—A Hitherto Unexplored Field of the Dichotomy between CRP and Ferritin Expression in Patients with Hyper Inflammation and Failure to Increase Ferritin"

_ijms, 2023, doi:10.3390/ijms241411350_

Round 1

Reviewer 1 Report

Dear authors,

the paper should be expanded with the introduction of other hematological values ​​to exclude anemia or other genetic diseases. If you have hemoglobin, iron and transferrin saturation values, you must enter them in the table. Most of the patients are women with an average age of 80 years. More data on these patients is needed.

In the classification I do not understand patients with COVID-19 if they are included in the series for ferritin and PCR. Table 3 shows the medians at the top, I would move them after the data because that position blends into the underlying data. Also still in table 3 I don't understand the percentages, explain yourself better.

Author Response

We thank the reviewer for his comments and the opportunity to improve our manuscript by addressing the comments.

Comment 1: The paper should be expanded with the introduction of other hematological values to exclude anemia or other genetic diseases. If you have hemoglobin, iron and transferrin saturation values, you must enter them in a table.

Response 1: We entered hematological values in a table 3. Furthermore, we added a comment in the discussion section regarding anemia and the hematological values of the cohort.

Comment 2: Most of the patients are women with an average age of 80 years. More data on these patients is needed.

Response 2: We added an explanation towards the end of the discussion section regarding the high percentage of elderly women in the cohort.

Comment 3: In the classification I do not understand patients with COVID-19 if they are included in this series for ferritin and CRP

Response 3: COVID-19 patients were included in the cohort and an explanation was added at the end of the discussion section.

Comment 4: Table 3 shows the medians at the top, I would move them after the data, because that position blends into the underlying data. Also still in table 3 I don’t understand the percentages, explain yourself better.

Response 4: We changed the table according to the reviewer’s comments. The medians were removed from the top and we changed the percentages to interquartile ranges.

Reviewer 2 Report

The work entitled “Normoferremia in Patients with Acute Bacterial Infections; A Hitherto Unexplored Field of Dichotomy between CRP and Ferritin Expression in Patients with Hyper Inflammation and Failure to Increase Ferritin” reports on the relationship between acute bacterial infections and ferritin using medical records as the methodology to assess the relationship. Generally, the work is of interest and deals with a pertinent subject for the medical community.

The introduction is very well put together and the novelty is clear. The methodology is detailed enough and the results are very well presented. The only issue with this manuscript is with the English writing, which is a bit deficient and some mistakes can be found, and with the lack of criticism from the literature in the discussion. The authors should try and corroborate their findings with other similar works or demonstrate in which way their work surpasses former studies. Regardless, the work is of quality and should be considered for publication after minor revision.

There are many grammar mistakes that require the authors' attention.

Author Response

We thank the reviewer for his comments and the opportunity to improve our manuscript by addressing the comments.

Comment 1: The only issue with this manuscript is with the English writing, which is a bit deficient and some mistakes can be found.

Response 1: The English writing of the manuscript had been thoroughly edited according to the reviewer’s comment.

Comment 2: The only issue with this manuscript is with the English…and with the lack of criticism from the literature in the discussion. The authors should try and corroborate their findings with other similar works or demonstrate in which way their work surpasses former studies.

Response 2: To the best of our knowledge this is the first study to explore the existence of normoferremia in acute infection and inflammation. It is well known that ferritin is acute phase protein. The novelty of our observation is the lack of ferritin increment in the presence of a major inflammatory response as documented by very high CRP concentration. We added a relevant remark to the discussion.

Round 2

Reviewer 1 Report

Dear Authors,

now the paper thus modified provides a better understanding of the patients' iron status. The patients, as you reported in the paper, are all or almost all anemic with very low saturation %. So normoferritinemia could be affected by this. It would have been nice to add a table with the ferritin and crp values ​​after the inflammatory state of the same patients to verify the true state of their ferritin. Talking about free iron released by ferritin in the discussion seems a bit risky to me. With such low saturation %, if there are no other causes, it is difficult to find free iron in the circulation. You would have to enter the value of NTBI in order to make such a guess. The discussion is a bit poor.

Author Response

We thank the reviewer for his comments and the opportunity to improve our manuscript by addressing the comments.

Comment 1:

Now the paper thus modified provides a better understanding of the patients' iron status. The patients, as you reported in the paper, are all or almost all anemic with very low saturation %. So normoferritinemia could be affected by this. It would have been nice to add a table with the ferritin and crp values ​​after the inflammatory state of the same patients to verify the true state of their ferritin.

Response 1:

We agree with the reviewer’s comment and we would have liked to add a table with the ferritin and crp values ​​after the inflammatory state of the same patients to verify the true state of their ferritin. However, unfortunately we do not have these values and we added it as a limitation of our study at the end of the discussion.

Comment 2:

Talking about free iron released by ferritin in the discussion seems a bit risky to me. With such low saturation %, if there are no other causes, it is difficult to find free iron in the circulation. You would have to enter the value of NTBI in order to make such a guess. The discussion is a bit poor.

Response 2:

We agree with the reviewer’s comment regarding the difficulty to find free iron in the circulation in a state of low saturation %, therefore we removed the relevant sentence from the discussion.